The quality and reliability of patient education regarding sound therapy videos for tinnitus on YouTube

Huang Chao
Lan Hongli
Jiang Fan
Huang Yu
Lai Dan lz_ld@126.com
Department of Otolaryngology Head and Neck Surgery, the Affiliated Hospital of Southwest Medical University , Luzhou, Sichuan , China
Okpala Charles
Electronic publication date: 2024 Feb 1
Publication date: 2024
Volume: 12
Electronic Location ID: e16846
Received 2023 Jun 22; Accepted 2024 Jan 7
Copyright: © 2024 Huang et al.
Copyright year: 2024
Copyright holder: Huang et al.
License: This is an open access article distributed under the terms of the Creative Commons Attribution License, which permits unrestricted use, distribution, reproduction and adaptation in any medium and for any purpose provided that it is properly attributed. For attribution, the original author(s), title, publication source (PeerJ) and either DOI or URL of the article must be cited.
License URL: https://creativecommons.org/licenses/by/4.0/

Keywords: Tinnitus, Sound therapy, Patient education, Medical informatic, Public health, Otology

Funding: Doctoral Research Initiation Fund of Affiliated Hospital of Southwest Medical University 19028 This study was supported and funded by the Doctoral Research Initiation Fund of Affiliated Hospital of Southwest Medical University (Grant No. 19028). The funders had no role in study design, data collection and analysis, decision to publish, or preparation of the manuscript.

==============================
Background

Numerous online videos are available on sound therapy as a treatment modality for tinnitus, but it is uncertain if these videos are adequate for patient education. This study aims to evaluate the quality and reliability of tinnitus sound therapy videos on YouTube for patient education.

Methods

YouTube videos were searched using keywords related to “tinnitus sound therapy”. The top 100 videos were analyzed after excluding those were repetitive, irrelevant, less than 3 min, or not in English. After categorising the videos based on their authorship and content, the video power index (VPI) was relied to determine their popularity. The DISCERN questionnaire (DISCERN), the Global Quality Score (GQS), the Journal of the American Medical Association benchmark criteria (JAMA), and the Patient Education Materials Assessment Tool (PEMAT) were utilized to evaluate the quality, transparency, and patient education.

Results

Over half (56%) of the videos were published by professional organizations. A total of 93% of them contained sound only. Only 17% followed the recommendations of the Clinical Management of Tinnitus Guidelines, and 3% provided literature referenced by the video. A variety types of sound were used, among which music accounting for 35%. The videos were highly popular with an average views of 7,335,003.28 ± 24,174,764.02 and an average VPI of 4,610.33 ± 11,531.10. However, their quality was poor (the median scores: 38/80 for DISCERN, 2/5 for GQS, 1/4 for JAMA, and 50%/100% for PEMAT). There was a negative correlation between the popularity of the videos and their quality, indicated by PEMAT: −0.207, DISCERN: −0.307, GQS: −0.302, and JAMA: −0.233. Several dimensions of the videos require improvement, especially actionability, treatment options, and transparency with lacks of 100%, 63%, and 75% respectively.

Conclusion

The tinnitus sound therapy videos available on YouTube exhibit low quality. Nevertheless, they also hold potential for health education if refined and utilized suitably.

Introduction

Tinnitus is defined as an auditory sensation without external sound (Lv et al., 2016). Around 14% of adults globally report experiencing phantom sensations, out of which 2% suffer from severe cases. Moreover, an estimated 1% of new patients visiting healthcare providers every year experience such sensations (Jarach et al., 2022). There are several associated risk factors for tinnitus, including prolonged noise exposure, head and neck injuries, hyperlipidemia, osteoarthritis, asthma, and depression (Kim et al., 2015). About 50% of individuals diagnosed with hearing loss also report experiencing tinnitus, which is commonly associated with a reduction in hearing sensitivity (Khan et al., 2021). Bothersome tinnitus may also occur alongside hyperacusis, which is a condition characterized by an atypical reaction to everyday sounds that may ultimately result in mental distress and hinder one’s ability to engage in professional, recreational, and social pursuits (Aazh et al., 2018; Aazh & Salvi, 2020). Individuals with bothersome tinnitus may experience emotional stress, which could potentially lead to changes in their blood pressure, pulse, and metabolism of sugar, fat, and protein. Furthermore, this stress may suppress appetite and disrupt sleep, potentially leading to the development of depression and anxiety (Mazurek, Boecking & Brueggemann, 2019; Shore, Roberts & Langguth, 2016; Mazurek, Szczepek & Hebert, 2015).

Treatment strategies for tinnitus mainly aim to reduce the distress caused by tinnitus, including sound therapy (Wang et al., 2020). Tinnitus sound therapy refers to the use of sound to modify an individual’s perception or reaction to tinnitus. This technique is commonly used in many tinnitus management programs, such as those that involve hearing aids and sound generators (Tunkel et al., 2014; Sereda et al., 2019). Following the introduction of tinnitus mask therapy in 1975, researchers have conducted numerous studies to evaluate the effectiveness of different sound stimuli for managing tinnitus. Such stimuli include broadband noise, narrower bands of noise and tone bursts, music, and natural sounds (Pienkowski, 2019). The specific mechanism behind the effectiveness of sound therapy for tinnitus may be that it enhances hearing, enabling the brain to focus on meaningful external sounds while suppressing or even eliminating the perception of tinnitus (Osuji, 2021). Recent meta-analyses have confirmed that different sound therapies can significantly reduce tinnitus symptoms and enhance quality of life (Liu et al., 2021). This effectiveness is believed to result from a sensory processing approach that involves the masking, more efficient gating, residual inhibition, desynchronization, peripheral re-entry reversal of abnormal gain, and lateral inhibition (Searchfield, 2021). A systematic review found that approximately one-third of participants with tinnitus experienced benefits from maskers alone, without the use of counseling. Additionally, a significant percentage of participants reported improvements in various areas, including mood (75%), sleep (55.6%), attention (50%), and hearing (37.5%) (Tyler et al., 2020).

At present, tinnitus sound therapy is not limited to specialized medical institutions, but also is gradually being adapted for home use and electronic mobile devices. This development aims to increase convenience for patients and make therapy more accessible (Sereda et al., 2019). A recent study indicates that over 50% of patients worldwide prefer to seek information about their disease through the internet (Erdem & Karaca, 2018). Individuals experiencing tinnitus symptoms can explore sound therapy through mobile applications or online videos available on various platforms for information on managing and treating their condition (Mehdi et al., 2020). The influence of video sites is noticeable. For instance, YouTube is the most popular video platform in the world, with an average of 400 h of video uploaded on it every minute. According to the statistics, more than 70% of YouTube’s views originate from mobile electronic devices (YouTube, 2022). Despite being primarily known as an entertainment platform, YouTube has also become a popular source of health-related information (Ranade et al., 2020), including sound therapy for tinnitus.

It is crucial for health-related information to facilitate patients’ comprehension of the disease’s natural progression. This understanding can equip them with the necessary abilities to manage both their medications and illness, allowing them to adjust their treatment regimen and sustain a good quality of life (Stenberg et al., 2018). Studies have shown that the information available online for patient education can play a significant role in shaping their understanding of the disease and affecting their decision-making. However, it is important to note that the complexity and inconsistency of such information can pose challenges for patients seeking reliable guidance (Hamers, Hibbard & Visser, 2010). According to research, YouTube health videos tend to receive more engagement from viewers when they incorporate personal anecdotes, inaccurate information, or non-evidence-based treatments (Langford & Loeb, 2019).

Although there is a vast number of tinnitus sound therapy videos available on YouTube, current research has not determined the extent to which these videos can assist clinical practice or if they contain inaccurate or inadequate information for diverse medical conditions. Thus, in this study, evaluation tools that are globally recognized will be employed to appraise the reliability and quality of videos.

Materials and Methods

Study design

This study employed a cross-sectional design to conduct a presentational analysis of publicly available YouTube videos.

Materials screening

Researchers deleted the search history and cleaned the cookies before searching to reduce the search-related bias. Videos with “tinnitus sound therapy”, “sound therapy”, “tinnitus treatment”, “tinnitus masking therapy” as the keywords up to August 1, 2021 were retrieved. The first 100 videos were collected for analysis (Ferhatoglu & Kudsioglu, 2020; Manchaiah et al., 2020; Yildiz & Toros, 2021; Morahan-Martin, 2004). To avoid duplication and irrelevant materials, only videos that were of sufficient length (more than 3 min), in English language, and specifically related to tinnitus were included in the study. The videos were classified into various categories, such as physicians, patients, professional organizations, personal media, and unidentified sources based on their publishers. In the present study, professional bodies are defined as online organizations, concentrating on sound therapy. Personal media, on the other hand, pertain to individuals who are distinct from institutions and patients, while the unknown category pertains to authors whose identities cannot be clearly determined.

Content analysis

The contents mentioned in the videos and their blurbs were analyzed. We classify the videos as lectures, sound only, experience sharing and other. Videos containing sound clips were categorized based on the type of sound utilized, such as broadband noise (BBN), narrowband noise, music, nature sounds, or mixed sounds. Moreover, an analysis was performed to ascertain whether the sound therapy methods employed in the videos complied with the clinical management guidelines established by Tunkel et al. (2014) (the patient received comprehensive information on the principles behind sound therapy, the necessary equipment, as well as the pros and cons), or if the literature cited was clearly indicated.

Popularity calculating

Erdem & Karaca (2018) first introduced the concept of video power index (VPI), which is primarily utilized to determine a video’s popularity based on the following formula: the number of views multiplied by likes divided by the number of days since uploaded, and then further multiplied by the sum of likes and dislikes. The VPI represents the play and like rates of a video over a specific duration, without any established upper or reference value. As a general rule, the higher the VPI value, the more popular the video is perceived to be.

Quality evaluation

DISCERN questionnaire

The instrument, which consists of three sections with 16 questions each, was developed by Charnock et al. (1999) and has been recognized internationally for its effectiveness in measuring quality of health information online, i.e., reliability, treatment choices, and overall rating. The questions in each section were rated on a scale of 1 to 5 based on the level of compliance observed. Full compliance was assigned five points, while non-compliance received one point, and partial compliance was rated between two and four points. The highest possible total score for each item was 80, while the lowest achievable score was 16 (Kartal & Kebudi, 2019).

Journal of American Medical Association benchmark criteria

The Journal of American Medical Association (JAMA) criteria were utilized to obtain transparency concerning whether the information given had fulfilled the criteria rated from 0–4, which are authorship, attribution, disclosure, and currency (Zhang, Sun & Xie, 2015). Authorship criterion requires providing authors’ and contributors’ affiliations and relevant credentials. Attribution criterion necessitates clear listing of references, sources for all content, copyright information, etc. Disclosure criterion mandates prominently disclosing web site ownership, as well as any sponsorship, advertising, underwriting, commercial funding arrangements or support, or potential conflicts of interest. Currency criterion entails indicating the dates of content posting and updating.

Global quality score

Langille et al. (2010) proposed a five-level categorization called global quality score (GQS) for evaluating the practicality of video materials. The levels range from poor quality with limited usefulness to patients (level 1 and 2) to good quality with coverage of important topics (level 4) and excellent quality and flow that are highly beneficial to patients (level 5) (Langille et al., 2010).

Patient education materials assessment tool

Shoemaker, Wolf & Brach (2014) developed an assessment tool for patient education that evaluates understandability and actionability of video content based on word choice and style, organization, layout and design, visual aids, and feasibility, using 17 items. The score is based on a one-point system for basic compliance, 0 point for less compliance, and N/A for no relevant information. To calculate the final score, divide the total obtained score (excluding N/A items) by the highest possible score and multiply by 100%. A higher score closer to 100% indicates better patient educability.

Scoring and statistical analysis

Two otology specialists independently conducted all the evaluations, and in case of discordant results, they held discussions to reach a consensus rather than averaging the scores. SPSS 26.0 software (IBM, Armonk, NY, USA) was utilized to perform the statistical analysis. The count data were expressed as percentile, while the quantitative data were presented as mean and standard deviation based on the results of the normality test. The Kolmogorov-Smirnov (K-S) test was used to determine whether the scores would be consistent with a theoretical distribution in the population. The Kruskal-Wallis test was used to compare the VPI and scores based on how the authors or content were grouped together. In addition, Spearman’s rank correlation analysis was applied to explore the correlation among VPI and the scoring systems. A two-tailed P < 0.05 was considered statistically significant.

Results

Video sources

Table 1 shows the distribution of authorship, type, adherence to Tunkel et al.’s (2014) guidelines, and compliance with other relevant literature standards for sound therapy videos. It was found that professional bodies were the main publishers of tinnitus sound therapy-based videos (56%), followed by personal media (27%), while only a small proportion of videos (approximately 17%) were attributed to sources other than the original author. The majority of videos (about 93%) only included sound, while the remaining types of videos had comparable shares (approximately 3%, 1%, and 3%, respectively). The “other” category accounted for only 1% of the videos because it included elements from the other three types. The minority of the videos met the requirements of the clinical management guidelines for tinnitus (17%), and only a very small proportion of videos provided references to literature (3%).

Table 1 Grouping and proportion of videos.

Items	Number (Percentage)	
Author (n = 100)			
	Physicians	7 (7%)	
Professional bodies	56 (56%)	
Patients	1 (1%)	
Personal media	27 (27%)	
Unknown	9 (9%)	
Content (n = 100)			
	Lecture	3 (3%)	
Sound only	93 (93%)	
Experience sharing	3 (3%)	
Other	1 (1%)	
Align with guideline (n = 100)	17 (17%)	
Mentioning literature (n = 100)	3 (3%)	

The videos that comprised solely of lectures accounted for a minimal percentage (3%) where sound clips had not been incorporated. Among the remaining videos (n = 97), music was found to be the most frequently employed stimulus (35%), and it was always produced by the publishers themselves. Other types of sound stimuli were also widely used and their least common was mixed sound (8%). In addition, broadband noise, narrowband noise, and natural sound accounted for 24%, 15%, and 18%, respectively.

Video popularity

The means for views, likes and VPI in the videos analyzed were 7,335,003.28, 47,707.70 and 4,610.33 respectively, which demonstrated the popularity of sound therapy resources compared to previous studies (Yildiz & Toros, 2021; Gokcen & Gumussuyu, 2019; Kunze et al., 2020). Table 2 presents the median values of play data along with the associated video performance indicator (VPI). Videos published by professional bodies had the highest number of views, likes, and VPI (987,820.50, 11,000.00, and 1,277.24, respectively). In contrast, VPIs for videos published by personal media and lecture-type videos were the lowest at 130.41 and 211.06, respectively. There was a significant difference in the median VPI scores among multiple groups according to author (H = 20.092, P = 0.000), while there was no significant difference in the median VPI scores according to the content (H = 2.071, P = 0.558).

Table 2 Video play data and VPI (shown as median ± interquartile range).

	Number of views	Thumbs up	Thumbs down	Days since uploaded	VPI	
Physicians	593,746.00 (±6,465,007.00)	5,281.00 (±98,205.00)	294.00 (±3,961.00)	791.00 (±1,380.00)	489.99 (±3,462.33)	
Professional bodies	987,820.50 (±6,147,350.00)	11,000.00 (±78,049.00)	602.00 (±4,300.00)	1,009.00 (±1,091.00)	1,277.24 (±5,415.41)	
Patients	484,964.00 (±0)	12,000.00 (±0)	1,069.00 (±0)	569.00 (±0)	782.59 (±0)	
Personal media	112,562.00 (±685,987.00)	792.00 (±6,336.00)	96.00 (±586.00)	548.00 (±1,125.00)	130.41 (±847.79)	
Unknown	308,239.00 (±377,989.00)	1,299.00 (±2,408.00)	121.00 (±138.00)	1,975.00 (±1,772.00)	141.36 (±154.51)	
Lectures	102,800.00 (±/)	1,795.00 (±/)	102.00 (±/)	521.00 (±/)	211.06 (±/)	
Sound only	562,682.00 (±2,543,910.00)	4,541.00 (±31,143.00)	356.00 (±1,536.00)	991.00 (±1,210.00)	520.00 (±2,989.80)	
Experience sharing	484,964.00 (±/)	8,560.00 (±/)	562.00 (±/)	373.00 (±/)	1,089.06 (/)	
Others	1,523,345.00 (±0)	20,000.00 (±0)	873.00 (±0)	548.00 (±0)	2,663.56 (±0)	
Note:

Kolmogorov-Smirnov (K-S) test, all P < 0.01 (number of views: 0.000, thumbs up: 0.000, thumbs down: 0.000, upload days: 0.001, VPI: 0.000). Kruskal-Wallis H test between multiple groups, authors: number of views: P = 0.004, thumbs up: P = 0.000, thumbs down: P = 0.001, upload days: P = 0.255, VPI: 0.000; content: number of views: P = 0.406, thumbs up: P = 0.610, thumbs down: P = 0.507, upload days: P = 0.239, VPI: 0.558. VPI, video power index.

Video quality

Figure 1 shows the distribution of the PAMET, DISCERN, GQS, and JAMA scores for the sound therapy videos. It was revealed that the heterogeneous distribution of PAMET and DISCERN scores, implies great variations in patient education and reliability for tinnitus sound therapy videos. Furthermore, it is shown that GQS and JAMA scores for sound therapy videos are mostly concentrated in the lower score brackets, indicating significant deficiencies in practicality and transparency.

Figure 1 Frequency distribution of PEMAT, DISCERN, GQS, and JAMA scores.

PEMAT, the patient education materials assessment tool; DISCERN, DISCERN questionnaire score; JAMA, the Journal of American Medical Association score; GQS, the global quality score.

The median, minimum, and maximum scores of the PAMET, DISCERN, GQS, and JAMA are shown in Table 3, and the statistics presented are segregated based on the author and content. Videos published by physicians and lecture-type videos had the highest median scores (PEMAT: 67% and 73%; DISCERN: 51 and 54; GQS: 3 and 3; JAMA: 2 and 2), followed by professional bodies and sound-only (PEMAT: 55% and 50%; DISCERN: 38 and 38; GQS: 2 and 2; JAMA: 1 and 1). In addition, the highest scores were mainly from videos posted by physicians and professional bodies, as well as from videos with lecture type and sound only. The differences in PEMAT, DISCERN, GQS, and JAMA scores were significant among multiple groups according to author and content (author: P = 0.014, 0.013, 0.020, and 0.009, respectively; content: P = 0.018, 0.005, 0.012, and 0.025, respectively).

Table 3 Scores for different groups (median (minimum, maximum)).

		PEMAT	DISCERN	GQS	JAMA	
Total		50% (27%, 73%)	38 (27, 58)	2 (1, 3)	1 (0, 3)	
Author	Physician	67% (45%, 73%)	51 (29, 58)	3 (2, 3)	2 (1, 2)	
	Professional body	55% (33%, 73%)	38 (28, 53)	2 (1, 4)	1 (0, 3)	
	Patient	45% (/, /)	34 (/, /)	1 (/, /)	0 (/, /)	
	Personal media	45% (27%, 64%)	38 (27, 50)	2 (1, 3)	1 (0, 2)	
	Unknown	45% (36%, 64%)	34 (30, 48)	2 (1, 3)	1 (0, 2)	
Content	Lecture	73% (67%, 73%)	54 (50, 58)	3 (3, 3)	2 (2, 2)	
	Sound only	50% (27%, 73%)	38 (27, 53)	2 (1, 4)	1 (0, 3)	
	Experience sharing	45% (36%, 45%)	32 (29, 34)	1 (1, 2)	0 (0, 1)	
	Other	46% (/, /)	32 (/, /)	1 (/, /)	0 (/, /)	
Note:

One-sample Kolmogorov-Smirnov test. DISCERN: P = 0.002; PEMAT, JAMA and GQS: P = 0.000. DISCERN, DISCERN questionnaire score; JAMA score, the Journal of American Medical Association score; GQS, global quality score.

Four instruments were employed to evaluate the understandability, actionability, reliability, treatment choices, overall quality, transparency, and practicability of the material. Figure 2 shows the gap between the actual score of the material and the ideal perfect score related to the seven dimensions mentioned above. As the data did not conform to a normal distribution, we calculated the median scores for all included videos in each dimension to present the actual score (one-sample Kolmogorov-Smirnov test, P < 0.05). The outcome was demonstrated as the black bar, while the complete score for each quality aspect was shown as white bars. It was evident that noticeable opportunities for enhancement were existed in all of these domains, particularly in the domains of actionability, treatment choices, and transparency (100%, 63%, and 75%, respectively).

Figure 2 The gap between actual scores and the ideal perfect scores.

(A) Understandability, (B) actionability, (C) reliability, (D) treatment choices, (E) overall quality, (F) transparency, (G) practicability one-sample Kolmogorov-Smirnov test: understandability, actionability, reliability, overall quality, transparency and practicability: P = 0.000; treatment choices: P = 0.001.

Correlation analysis of the VPI with quality

Table 4 reveals the correlations between the VPI and scoring instruments (r is indicative of the correlation coefficient). It was found that there were negative correlations between VPI and each scoring instrument (PEMAT: r = −0.207, P = 0.039; DISCERN: r = −0.307, P = 0.002; GQS: r = −0.302, P = 0.002; JAMA: r = −0.233, P = 0.020). There is a correlation between a video’s popularity and its likelihood of having lower quality. In addition, there were positive correlations between the each two scoring tools, suggesting that the low quality of the videos was reflected in all aspects rather than in a single aspect.

Table 4 Assessment of the relationship between VPI and scores.

Items		VPI	PEMAT	DISCERN	GQS	JAMA	
VPI	R		−0.207*	−0.307**	−0.302**	−0.233*	
	P		0.039	0.002	0.002	0.020	
PEMAT	R	−0.207*		0.885**	0.727**	0.742**	
	P	0.039		0.000	0.000	0.000	
DISCERN	R	−0.307**	0.885**		0.805**	0.774**	
	P	0.002	0.000		0.000	0.000	
GQS	R	−0.302**	0.727**	0.805**		0.734**	
	P	0.002	0.000	0.000		0.000	
JAMA	R	−0.233*	0.742**	0.774**	0.734**	1.000	
	P	0.020	0.000	0.000	0.000		
Note:

Spearman’s rho, r: correlation coefficient.

** Correlation is significant at the 0.01 level (two-tailed).

* Correlation is significant at the 0.05 level (two-tailed).

VPI, video power index; PEMAT, the patient education materials assessment tool; DISCERN, DISCERN questionnaire score; JAMA score, the Journal of American Medical Association score; GQS, global quality score.

Discussion

In the present research, it was found that tinnitus sound therapy-based videos had a very high viewership and the most popular videos were respectively professional bodies posted as well as sound only videos. The reason why individuals favor videos produced by professional organizations is probably their trust in established organizations. Besides, considering the duration of being online, the reason for the popularity of sound-only videos may be that sound-only videos were among the earliest to be published, and they possibly had an impact on the subsequent release of sound therapy videos. However, it is essential to note that popularity and quality are not synonymous. If an online health video fails to deliver accurate and informative content despite high views, viewers may become hesitant, skeptical, and resistant towards clinically supportive information (Morahan-Martin, 2004; Wong & Levi, 2017).

Thus, several internationally recognized scoring criteria were herein utilized: DISCERN questionnaire, GOS, JAMA-benchmark, and PEMAT, which concentrated on the quality, practicability, transparency, and patient educability of the material. It was found that the quality and patient educability of tinnitus sound therapy-based videos on YouTube are generally low. Besides, it was revealed that physician-sourced and lecture videos had higher quality and patient education, while they only accounted for 7% and 3% of the videos. The factors contributing to the substandard quality of videos may be diverse. Firstly, a considerable proportion of the videos fail to comply with clinical management guidelines for tinnitus, which involves informing patients about potential outcomes and associated costs (such as emotional and financial costs), according to reference (Tunkel et al., 2014), and rarely make explicit references to the literature. Secondly, some authors who create sound therapy videos lack professional expertise and thus are limited in their ability to produce therapeutic sounds. As a result, they tend to produce pleasant sounds, but there is not enough assurance of efficacy for the tinnitus itself. Furthermore, most videos offer minimal transparency, providing only sound clips without sufficient details such as the source, mechanism, commercial use, conflict of interest, precautions, and instructions on how to use them. Moreover, many sound therapy videos typically emphasize the benefits of this treatment approach, while tend to overlook its drawbacks, including inconvenience, cost, and dissatisfaction (Tunkel et al., 2014). Additionally, these videos mainly fall short of providing information on alternative treatments or advising patients to seek medical help from clinicians when a range of sound stimuli have been tried without success. Consequently, tinnitus patients may experience difficulties in accessing accurate information and appropriate treatment options from these videos.

The findings of the present study revealed that music stimuli were the most commonly used sound source in these videos, followed by narrowband noise (NB), broadband noise (BBN), and nature sounds. However, there are certain issues associated with these sound stimuli. Firstly, most of the published stimuli are created by the publishers themselves without any mention of the principles or mechanisms involved, making it difficult to assess their reliability. Besides, the effectiveness of different types of sound stimulation has been found to vary across different studies and evaluation criteria (Hoare et al., 2013). While BBN and NB noise have been recognized as more effective than other types of sound sources, including natural sounds, the efficacy of notch-filter music has shown considerable variation (Pienkowski, 2019), several publishers still prefer to use musical stimuli. Most importantly, a large number of videos neglect to provide patients with sufficient information regarding the use of sound stimulation, such as the selection of headphones or hearing aids, intensity levels (mixing point BBN, partial masking, or full masking), tinnitus pitch (which may not be matched by viewers of online videos), etc. As a result, significant variations in efficacy between online and clinical sound therapy may arise.

Additionally, it was found that there were negative correlations between VPI and quality ratings. This may be related to the fact that the tinnitus sound therapy-based videos posted by doctors are mostly lectures, which make patients bored or reluctant to spend time watching them, whereas videos from other sources are mainly audio clips that stimulate patients to attempt the therapy more quickly. In addition, patients mainly select videos based on high play count, high like count, and short video length, rather than a thorough evaluation of the content. Consequently, the authenticity and reliability of the videos are typically overlooked. This trend further exacerbates the popularity of already popular videos, while leaving lesser-known videos unnoticed. Additionally, as publishers upload more videos to their channels, they gain more fans, resulting in their later videos receiving more views. However, the quality of videos themselves does not necessarily improve. In light of the abovementioned findings regarding sound stimuli, it is crucial for publishers to prioritize factors, such as treatment choice, transparency, and actionability over the novelty and appeal of video content. Hence, they can avoid straying away from the core purpose of patient education.

To date, numerous studies have been conducted to assess the quality of online health information available on YouTube (Erdem & Karaca, 2018; YouTube, 2022; Ferhatoglu & Kudsioglu, 2020; Manchaiah et al., 2020; Kunze et al., 2020; Stellefson et al., 2014). This study’s findings are consistent with previous research, suggesting that there is a large variation in the quality and patient educability of relevant videos on YouTube. It was also found that videos uploaded by professors tend to be more educational, whereas have fewer views. This trend may indicate that publishers prioritize video popularity over educational value, opting for simpler formats or content to increase likes. However, this approach undermines the videos’ potential as educational resources. Patients mainly report finding inadequate or inconsistent information online, leading to changes in their treatment strategies, questions for clinicians, or even a shift in their health maintenance approach (Kunze et al., 2020). Thus, when patients inquire about tinnitus sound therapy, healthcare professionals should assist them in selecting high-quality videos. Educating patients to comprehend the information provided in videos with greater care and avoiding the reliance on metrics such as the number of views, likes, comments, or the novelty of the content, will result in a more accurate and comprehensive understanding of sound therapy. This will reduce unnecessary expenditures of time and cost for patients.

The present study differs from previous research as we concentrated specifically on sound therapy and analyzed videos without verbal narrative as a single audio segment. Additionally, internationally recognized evaluation tools were utilized to assess the quality of videos, while characteristics of sound therapy were also incorporated to provide a more comprehensive and accurate analysis of the sound stimuli used. However, this study’s limitation is that it only examined sound therapy-based videos from a limited number of websites and software on smart devices. Future studies should include more sources. Furthermore, as there are no standardized or feasible criteria for evaluating the effectiveness of sound therapy-based videos for tinnitus, future studies are suggested to incorporate patients’ comments into the quality assessment process to improve the results.

Conclusions

After evaluating online information on tinnitus sound therapy, it was revealed that it attracted a significant number of YouTube viewers, while lacks quality and patient education. As a result, the openness of the video platform may lead to misleading patients with inaccurate information. However, it is essential to acknowledge that the educational potential of online video platforms cannot be ignored, and they can be an important tool for health education to the public.

We are appreciative to authors who defined the scoring criteria, publishers who uploaded the related videos, and Etopsci for its linguistic assistance during the preparation of this manuscript.

Additional Information and Declarations

Competing Interests

Author Contributions

Data Availability

The authors declare that they have no competing interests.

Chao Huang conceived and designed the experiments, performed the experiments, analyzed the data, prepared figures and/or tables, authored or reviewed drafts of the article, and approved the final draft.

Hongli Lan performed the experiments, prepared figures and/or tables, and approved the final draft.

Fan Jiang performed the experiments, prepared figures and/or tables, and approved the final draft.

Yu Huang performed the experiments, prepared figures and/or tables, and approved the final draft.

Dan Lai conceived and designed the experiments, authored or reviewed drafts of the article, and approved the final draft.

The following information was supplied regarding data availability:

The data that support the findings of this study are available in Science Data Bank: Chao Huang, Dan Lai. The quality and reliability of patient education regarding sound therapy videos for tinnitus on YouTube [DS/OL]. V2. Science Data Bank, 2023 [2024-01-10]. https://cstr.cn/31253.11.sciencedb.13440. CSTR:31253.11.sciencedb.13440.

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
