# Peer review of "The quality and reliability of patient education regarding sound therapy videos for tinnitus on YouTube"

_PeerJ, doi:10.7717/peerj.16846_

## Round 0.1 · original submission · Major Revisions

Reviewers have attended to your work. Please kindly address the comments raised and pay close attention to providing strong details, not only in the revised manuscript but also in your reply. Thank you.

**Language Note:** PeerJ staff have identified that the English language needs to be improved. When you prepare your next revision, please either (i) have a colleague who is proficient in English and familiar with the subject matter review your manuscript, or (ii) contact a professional editing service to review your manuscript. PeerJ can provide language editing services - you can contact us at [email protected] for pricing (be sure to provide your manuscript number and title). – PeerJ Staff

·

Basic reporting

The English language is unambiguous that an international audience can clearly understand the text.
The Intro & background of the text stated clearly why do this research and explained the meaningful of this research. The literature well referenced.
The Structure conforms to PeerJ standards.
Figures are well labelled.

Experimental design

Research question well defined and meaningful for clinic.
Methods described with sufficient detail and information to replicate.
Conclusions are well stated, linked to original research question & limited to
supporting results.

Validity of the findings

All underlying data have been provided.The tables and figures are provided adequate data.

Additional comments

Please provide the reason why choose the first 100 videos to analysis.Does it only 100 videos meets inclusion criteria?

Reviewer 2 ·

Basic reporting

Dear authors,

this is a well-written and well-structured manuscript on a highly relevant topic with a beautiful mixture of innovative methods used to analyse the data.

The introduction provides a sufficient background on the topic. However, I am missing an overview on the effectiveness of sound therapy in general (mixed findings). Also, you could differentiate a bit more between the different sound therapeutic approaches such as masking, residual inhibition etc., since sound therapy is a very broad concept.

The results are well-suited to the defined research question.

Experimental design

The research question was well defined and is plausibly filling a research gap.

I really liked the selection of different and innovative methods (such as the video power index) where quantitative and qualitative data were combined.

The methods are structured and described sufficient. However, I have a few specific comments on this section:

Row 143: Please define sufficient length.
Row 156: Please define these guidelines shortly.
Row 169: Quality of what? Please explain what exactly the DISCERN questionnaire is measuring.
Row 189: Check citation.

Validity of the findings

The results are very well structured and presented in Figures and Tables. I cannot see if the raw data is shared somewhere, please consider mentioning.

Specific comments on the result section:

210: Table 1 shows the distribution of authorship etc. for sound therapy videos, not the distribution of videos.
233: Do you mean the duration of being online?
233-235: Consider moving this to the discussion.
Fig. 1 and its description does no contain relevant information in my opinion, please describe this result more substantially or consider removing it.
267: Please describe what exactly was tested here.
Fig. 2: What is the actual score? Mean of all videos or results of one single video?


The discussion needs some revision. First of all, some statements are not driven by the data or need some more explanation, such as:

282: In the present research, it was found that tinnitus sound therapy-based videos had a very wide audience
284: The reason why individuals perceive professionally produced videos as more credible
374: After evaluating online information on tinnitus sound therapy, it was revealed that it is of high popularity among YouTube viewers

Also, some general statements are, to my knowledge, not backed by the literature and if so, need some good reference:

302: sound therapy may be ineffective for patients with non-bothersome or non-persistent tinnitus
352: 1) The efficacy of sound therapy may be limited in patients with non-bothersome or mild tinnitus (I am not sure if this is true. Sound therapy can also help patients with mild tinnitus as far as I know.)

In general, please be careful with your statements on effectiveness of those videos that you evaluated, because you have not testet it.

Specific comments on the discussion section:

300: Pleasing sounds can have a therapeutic effect, too.
355: Statement 4: How should patients evaluate those factors? This is not clinically helpful.

---

## Round 0.2 · Minor Revisions

Please, the reviewers have considered your work favorably. But some additional work is required to improve the quality. Kindly address the concerns raised, and where applicable, provide the direction for future studies.

·

Basic reporting

The language is clear. The article provide enough literature references and background.The structure of the article is consistent with "standard sections".

Experimental design

The experimental design is reasonable.

Validity of the findings

no comment

Additional comments

I'm glad to review the article. The reply to my question is satisfied.

Reviewer 2 ·

Basic reporting

Line 87-88: formatting issue

Experimental design

Line 171: DISCERN: Please make clear, that this instrument is used for validating patient information/material online.

Validity of the findings

Line 233: Could you give a reference for the video popularity index? It is hard to imagine what this high numbers could mean.

Additional comments

The manuscript has been improved substantially through the last revision, congrats! I have some minor comments left.

---

## Round 0.3 · accepted · Accept

Authors, thank you very much for the effort you have put in to address all the concerns raised by the reviewers, which have helped elevate the quality of your work. It is now acceptable for publication. PeerJ appreciates your scholarly work and looks forward to your future contributions. Congratulations :)

Reviewer 2 ·

Basic reporting

The authors have responded satisfactorily to all comments. There is only one typo left, in line 174: chioces -> choices.

I have no other comments. Congratulations to your work.

Experimental design

-

Validity of the findings

-

Additional comments

-